# Effect of a Contrast Agent on Bone Mineral Density Measurement in the Spine and Hip Using QCT-Conversion Factor Recommendation

**DOI:** 10.3390/jcm12041456

**Published:** 2023-02-12

**Authors:** Katharina Jäckle, Sophia Lüken, Paul Jonathan Roch, Friederike Sophie Klockner, Max Reinhold, Marc-Pascal Meier, Thelonius Hawellek, Wolfgang Lehmann, Lukas Weiser

**Affiliations:** 1Department of Trauma Surgery, Orthopaedics and Plastic Surgery, University Medical Center Göttingen, Robert-Koch Str. 40, 37075 Göttingen, Germany; 2Department of Radiology, Charité University Medical Center Berlin, Campus Benjamin Franklin, Hindenburgdamm 30, 12203 Berlin, Germany

**Keywords:** osteoporosis, contrast agent, spine, QCT, BMD

## Abstract

Background: Osteoporosis causes an increased fracture risk. Clinically, osteoporosis is diagnosed late, usually after the first fracture occurs. This emphasizes the need for an early diagnosis of osteoporosis. However, computed tomography (CT) as routinely used for polytrauma scans cannot be used in the form of quantitative computed tomography (QCT) diagnosis because QCT can only be applied natively, i.e., without any contrast agent application. Here, we tested whether and how contrast agent application could be used for bone densitometry measurements. Methods: Bone mineral density (BMD) was determined by QCT in the spine region of patients with and without the contrast agent Imeron 350. Corresponding scans were performed in the hip region to evaluate possible location-specific differences. Results: Measurements with and without contrast agent administration between spine and hip bones indicate that the corresponding BMD values were reproducibly different between spine and hips, indicating that Imeron 350 application has a location-specific effect. We determined location-specific conversion factors that allow us then to determine the BMD values relevant for osteoporosis diagnosis. Conclusions: Results show that contrast administration cannot be used directly for CT diagnostics because the agent significantly alters BMD values. However, location-specific conversion factors can be established, which are likely to depend on additional parameters such as the weight and corresponding BMI of the patient.

## 1. Introduction

Osteoporosis is a metabolic bone disease that affects in particular elderly patients of all genders. It is characterized by an increased risk of bone fractures, usually due to impaired bone remodeling and the resulting pathological bone microarchitecture [1]. This brief characterization is reflected in the Health Consensus Conference definition of osteoporosis [1], which defines osteoporosis as a skeletal disease characterized by decreased bone strength and an increased risk of fracture. Diagnosis of osteoporosis is usually very late, as the disease is clinically silent up until the first fracture occurs. Due to the demographic development, several studies predict an enormous increase in osteoporosis and resulting fractures in the coming decades [2,3]. The most common osteoporotic fractures affect the vertebral body. However, such clinically diagnosed fractures are often asymptomatic, and only approximately 30% of them are recognized [4]. Late diagnosis delays the initiation of the osteoporotic therapy. Notably, a late or missing recognition of such osteoporotic fractures is problematic because they increase the risk of subsequent vertebral fractures up to ten-fold [5]. Based on this prognosis, early and reliable diagnostic of osteoporosis becomes an increasingly important topic for both preclinical and clinical research. Up to now, standard radiological procedures for osteoporosis diagnosis include dual-energy X-ray absorptiometry (DXA) to quantitatively determine bone mineral density for the diagnosis of osteopenia or osteoporosis and for monitoring the success of the follow-up therapy.

Currently, osteoporosis guidelines and treatment recommendations are mainly based on DXA diagnosis. Although QCT examination is not considered equivalent to DXA, some studies have demonstrated a significant correlation between DXA and BMD as determined by QCT [6]. QCT and DXA diagnostics are different methods to determine BMD. QCT analysis is a medical technique to measure the trabecular BMD. It uses a standard CT diagnostic with a calibration standard to convert Hounsfield Units (HU) of the CT image to bone mineral density values [7]. In our study, we explored whether and how this method, i.e., QCT, could be used in combination with a contrast agent to measure trabecular BMD because it is more frequently available for routine clinical practice [6]. We performed QCT in combination with a contrast agent (Imeron 350; Bracco Imaging Deutschland GmbH, Constance, Germany) to measure bone mineral density. QCT measurements were performed with and without the contrast agent in each patient. We found a difference between native CT and CT after the contrast agent administration. This difference can be compensated by a location-specific conversion factor that can be determined experimentally.

## 2. Materials and Methods

### 2.1. Study Design and Subjects

The present study was approved by the ethics committee of the Medical Center Göttingen (approval number: AN 6/5/20) in compliance with the Helsinki Declaration. After applying relevant inclusion criteria for a retrospective cohort study, we analyzed a total of 130 patients (see Figure 1; mean age: 55.12 ± 19.24) who had received a CT scan with and without contrast administration at a Level-1 trauma center of the university clinic Göttingen during a four-year period (March 2017 until December 2021). Of the selected patients, 78 (60%) were male and 52 (40%) were female, respectively. Either whole-body images or partial-body images including the spine and/or proximal femur were evaluated (n = 70; mean age: 57.86 ± 18.63; 31 females and 39 males), including only 3 patients that received CT for the proximal femur. Only patients were recruited for the present study when the native and contrast-enhanced images were taken at a time interval of no more than six months. Fractured vertebral bodies, vertebral bodies with foreign material inserted, and vertebral bodies with CT morphologic pathologic changes (especially malignancy-susceptible osteoblastic/osteolytic lesions and benign bone tumors) were excluded from the analysis. These CT datasets were used for QCT analyses (QCT Pro^®^, version 6.1, Mindways Software; Kiel, Germany) in the thoracic and lumbar spine region and the proximal femur to perform bone densitometry measurements. First, the vertebral body used for the analysis was selected. The CT image was then rotated in the axial, sagittal, and coronal planes so that the position of the vertebral body to be measured corresponded to the 90°-angled auxiliary lines. Auxiliary cross marks were placed in the center of the vertebral body to be measured. An oval region of interest (ROI) was placed in the axial plane so that it included as much of the trabecular bone as possible while excluding the orifice of the basivertebral vein or areas of cortical bone. The placement of the ROI was then checked in all three section planes (axial, sagittal, coronal) and re-adjusted if necessary. Subsequently, the BMD values were automatically calculated for the ROI by the supporting software. Analogous to the vertebral body measurement, an area for the measurements was defined between the most cranial layer above the highest point of the femoral head and the most caudal layer approx. 1 cm below the trochanter minor. The selected image section was then rotated identically to the vertebral body measurement in all three section planes. A reconstruction of both the volume and the surface area of the proximal femur was then automatically generated by the software. Three independent measurements of BMD were made and the mean values were calculated.

### 2.2. Contrast Agent Administration

Imeron 350 (Bracco Imaging Deutschland GmbH, Constance, Germany) was used as the contrast agent. It contains 714.4 mg Iomeprol/mL and an iodine content of 350 mg/mL. Depending on their body weight, patients received 90–120 mL of the contrast medium via an intravenous catheter using an automatic application system (injector) with a flow rate between 3 and 4.5 mL/s. Native images as well as the portal venous contrast phase images were evaluated. The portal venous phase was started at an excess of 120 HU in the ascending aorta with a delay of 45 s.

### 2.3. Statistics

Statistical analysis was performed using the Quantile-Quantile-plot to check for normal distribution. Due to the presence of normal distribution, measurements were compared using a paired t-test. A Bland-Altman plot was created to compare the measurements with and without contrast agent administration. Linear regression was calculated to predict the native values from the contrast-enhanced values. Linear regression was performed separately for the spine and proximal femur. A paired t-test was performed to compare BMD measurement differences between spine and femur. The condition of normal distribution was again tested using a Quantile-Quantile-plot. Differences between groups were tested for significance using an independent-samples t-test. The trabecular BMD measurement differences between the normal BMD, osteopenia, and osteoporosis groups were tested for significance using the Kruskal–Wallis test. The tests were two-sided and the significance level was set to alpha = 5%. Values with *p* < 0.025 were considered significant. All data are shown as mean value ± standard deviation. SPSS (Version 28, IBM, Chicago, IL, USA) was used for statistical analysis.

## 3. Results

### 3.1. Comparison of the BMD Values of Patients with and without Contrast Agent Administration

We evaluated the effect of contrast administration on trabecular BMD measurements of the spine and the proximal femur by QCT imaging. A total of 127 spines and 70 proximal femurs were examined. The mean value of BMD in whole spine (measurements in the thoracic and lumbar spine) was 122.92 ± 48.16 mg/cm^3^ in the absence of contrast agent administration, and 143.80 ± 46.40 mg/cm^3^ after treatment with Imeron 350 contrast agent, respectively (see Figure 2 and Table 1).

In addition, we examined the bone density measurements of all vertebral bodies of the thoracic spine and the lumbar spine. Native BMD was found to be slightly increased in the whole thoracic spine with 129.97 ± 46.75 mg/cm^3^. Bone mineral density after contrast administration was 151.32 ± 45.72 mg/cm^3^, whereas the native BMD was 119.34 ± 45.65 mg/cm^3^ in the lumbar spine and 138.15 ± 43.35 mg/cm^3^ after contrast administration. However, the numerical difference between the vertebral bodies of the thoracic spine and the lumbar spine was not significant (*p* = 0.053) (see Table 1).

The mean BMD of proximal femur in native condition was 123.18 ± 37.48 mg/cm^3^ and 126.03 ± 36.66 mg/cm^3^ after contrast agent administration (see Figure 2 and Table 1). These differences were statistically significant for both spine (*p **** < 0.001) and proximal femur (*p **** < 0.001), indicating that there is a difference between the imaging measurements with and without contrast administration (see Figure 2). The differences in BMD measurements between contrast agent administration and without contrast agent administration are shown.

The Bland–Altman plot is a meaningful graphical tool for the comparison of two measurement methods and for assessing the agreement between two datasets. It shows that the two established methods of BMD determination without and after contrast agent agree well. The range of variation lies within the dashed lines, so that the accepted range of deviation is not exceeded (see Figure 3).

### 3.2. Effects of the Contrast Agent Amount on Bone Density Measurements in Spine and Proximal Femur

Figure 4 shows the difference in BMD between different amounts of contrast agent (minimum 90 mL to maximum 120 mL) for both spine and proximal femur measurements. The mean BMD values for spine were 23.56 ± 12.42 mg/cm^3^, 20.55 ± 16.31 mg/cm^3^, 18.83 ± 13.31 mg/cm^3^ and 18.01 ± 10.43 mg/cm^3^ for 90, 100, 110 and 120 mL of contrast medium, respectively (see Table 1). Corresponding mean values of proximal femur were 4.86 ± 5.22 mg/cm^3^ for 90 mL contrast administration, 3.06 ± 4.24 mg/cm^3^ for 100 mL administration, 2.01 ± 6.38 mg/cm^3^ for 110 mL and 4.76 ± 3.44 mg/cm^3^ for 120 mL. Most notably, the BMD differences observed either for spine or for proximal femur were not statistically significant.

It should be noted that the effect of the contrast agent is location specific as shown by a comparison of BMD measurements between spine and proximal femur (see Table 1). There is a statistically significant difference between the BMD values for both without and after contrast medium administration (*p *** < 0.003). The mean range of variation was approximately 20.88 mg/cm^3^ for the spine, but only approximately 2.85 mg/cm^3^ for the proximal femur (see Figure 2). We attribute this difference to the variable body weight of the patients and corresponding distribution of the contrast agent in their bodies. Figure 4 shows a combined diagram for both genders, since no significant difference between the male and female subjects has been detected.

### 3.3. Correlation of Body Mass Index and the Administration of Contrast Agents

Table 1 summarizes that there is a statistically significant negative correlation between the BMI and BMD after contrast agent administration (*p ** < 0.020). This implies that the higher a BMI value is, the more contrast agent is needed per kilogram of body weight. We attribute this finding to the fact that a higher body weight and the corresponding higher BMI of patients results in a different distribution of the contrast agent in their bodies. In addition, the magnified radiation hardening effects must be taken into account. In fact, the volume of the injected contrast agent is directly correlated with the body mass index. The body mass index (more precisely, the volume of tissue surrounding the examined object) determines the effect of radiation hardening which in turn leads to a reduction in HU.

### 3.4. Comparison between Contrast Agent Amount and Bone Mineral Quantity

Figure 5 shows that there are no significant differences between the BMDs of bone quality between normal people and patients suffering from osteopenia and osteoporosis patients with regard to the administered amount of the contrast agent. No significant differences in bone quality in terms of BMD values was detected when healthy bone (>120 mg/cm^3^) 23.14 ± 8.23 mg/cm^3^, bone with osteopenia (80–120 mg/cm^3^) 24.94 ± 8.96 mg/cm^3^ or bone with osteoporosis (<80 mg/cm^3^) 24.12 ± 10.46 mg/cm^3^ were compared with regard to the administered amount of contrast agent. There was also no correlation between contrast administration and BMD and bone quality.

The results show that the measurements of CT diagnostics after contrast administration and native CT diagnostics are correlated. Notably, contrast-enhanced CT diagnostics show better results than native CT diagnostics (Figure 6). Furthermore, the results revealed that the difference of calculated mean values after agent administration was higher for spine than for proximal femur. From these results a conversion factor could be obtained by linear regression analysis by the following calculation which was established to determine the corresponding native BMD from the measurements after contrast administration: ***BMD _native_ = (regression coefficient) × BMD _contrast agent application_ − mean value after contrast agent application.***

Using this general formula to determine the native BMD for spine and proximal femur suggest that one can determine the values for spine as follows: BMD _native_ = 0.986 × BMD _contrast agent application_ − 17.33, and for proximal femur BMD _native_ = 1.012 × BMD _contrast agent application_ − 4.34, respectively.

In summary, the data indicate that the CT data in combination with contrast agent administration cannot be used as initially recommended by the manufacturer [8], but require a transformation with a conversion factor that can be established in a location-specific manner.

## 4. Discussion

Guidelines for osteoporosis and its therapy recommendation are based on DXA-Scans. Some studies indicate a significant correlation between DXA and QCT, but they pointed out that DXA is more reliable than an QCT examination [9]. However, QCT is more accessible in clinical routine and a high number of patients have to undergo CT diagnostics during their hospital stay. Thus, it seemed reasonable to explore the possible use of QCT in addition to DXA for the diagnosis of osteoporosis and even osteopenia, because this technique is more precise than DXA since the bone measurements are less affected by surrounding structures such as calcifications of the aorta, facet joint arthrosis or the patient’s weight [9,10,11]. Using QCT diagnostics would therefore be more convenient for bone density measurements than DXA measurements. Regarding the measurement by QCT, a BMD value between 80 and 120 mg/cm^3^ characterizes osteopenia, whereas a BMD value below 80 mg/cm^3^ defines osteoporosis [9]. Thus, these diseases can be diagnosed and distinguished on the basis of BMD measurements.

Osteoporosis primarily affects the elderly. Unfortunately, the clinical diagnosis of osteoporosis is often only possible after a fracture occurs. Thus, an early diagnosis would be desirable to initiate therapeutic measures in time to prevent the occurrence of fractures and to thereby reduce morbidity and mortality. Up to now, X-ray-based imaging plays a key role in assessing fracture risk and monitoring osteoporosis. While DXA has been the most commonly used and recommended method for decades, QCT brings more recently another modality into play due to its three-dimensional advantages and the opportune use of routine CT scans [4,12]. However, the results of Löffler et al. [4] strongly suggest that existing CT scans can significantly improve quantitative osteoporosis imaging. This implies that, in addition to the established DXA technology, inclusion of further quantitative imaging methods such as QCT should be considered in future official osteoporosis guidelines.

Bauer et al. [13] also investigated the effect of contrast agent exposure on BMD in a much smaller patient population (spine: n = 40; hip: n = 21), focusing on vertebral bodies L1 to L3 including fractured vertebral bodies [13]. The authors concluded that contrast enhancement values of multidetector CT (MDCT) were on average 30.3% higher than those of QCT at the spine and 2.3% higher at the proximal femur (*p* < 0.05). Our results also showed a statistically significant difference after contrast administration at the spine and proximal hip (*p* *** < 0.003). In addition to such measurements of Bauer et al., we performed bone density measurements of all vertebral bodies of the thoracic spine and lumbar spine and found no significant differences concerning the QCT values of these vertebral bodies in the corresponding thoracic spine or lumbar spine sections.

Ziemlewicz et al. [14] demonstrated that the effect of intravenous contrast enhancement on area femoral neck BMD measurement using CT X-ray absorptiometry. Both unenhanced and contrast-enhanced CT series were compared. The authors concluded that for the purposes of opportunistic osteoporosis screening, contrast-enhanced abdominopelvic CT examinations are relatively equivalent to unenhanced CT and thus, they can be used to assess femoral neck BMD. In a recent review article, Kutleša et al. [15] pointed at possible effects of the administration of iodine-containing contrast agents on BMD estimated on CT. This review also assessed studies that used both CT measurement via volumetric quantitative BMD and CT attenuation in Hounsfield units. The review reports that in nearly all studies a significant increase in BMD values on the contrast-enhanced CT scans was observed, similar to what has been observed in our study, i.e., significant differences between BMD values of unenhanced and contrast-enhanced CTs were detected. The differences were more pronounced in the lumbar spine area than at the proximal thigh. The authors also note that the extent of the differences depends on age, gender, contrast administration and post-contrast imaging protocol.

Our results show a significant difference of the BMD values between native CT and CT after contrast agent application. Thus, CT diagnostics cannot directly be used for a correct BMD determination as recommended by the manufacturer [8]. We show that after calculating a conversion factor that accounts for the difference between the native CT values and the values obtained after contrast administration, QCT can be used in a rather convenient way. Surprisingly, this correlation factor was found to be location-specific, i.e., the influence of the contrast agent on the spine is higher than on the proximal femur, and there was a statistically significant difference between the BMD after contrast agent administration (*p* ** < 0.003) in different locations analyzed. Additionally, the contrast medium administration could be more accurately determined on the proximal femur than in the spine area. This finding indicates that due to the significantly smaller range of variations in the proximal femur, the determination of the diagnostically relevant BMD value should be performed on the proximal femur since measurements in this location appear to be more accurate than on the spine.

In summary, our study indicates that after administration of a contrast agent, QCT measurements can indeed be used instead of or in addition to DXA. In this case, QCT measurements cannot be taken directly but require transformation of the measured values by a compensation factor to determine the diagnostically relevant BMD value which is necessary for reliable determination of osteopenia or osteoporosis at an early stage, i.e., before first fractures occur. An advantage of QCT measurements to assess BMD is that this technique is more commonly used for routine clinical practice than DXA. Our study, and likewise the studies discussed in this section, show that this measure could significantly improve osteoporosis screening and help to establish the corresponding preventive medicine.

We should also note that our study has some important limitations since the number of analyzed patients is rather small. Thus, a general compensation factor could not yet be reliably established and has to wait until a much larger population of patients has been examined. Other limitations also include that QCT diagnostics were applied to individuals and not to groups. Thus, inter-individual differences may be included and could lead to a diagnostic bias. According to the position of the International Society of Clinical Densitometry, only BMD measurements of the lumbar vertebral bodies are required. However, we included measurements of the thoracic vertebral bodies because we felt that the data are still very limited. When using a synchronous phantom, the effect of increased radiation hardening effects was found to be influenced by BMI. We also realized that changes in BMD were due to peculiarities of the different vascular structure of the callus body and proximal femur and they should theoretically be well compensated for. However, our study shows that this is not the case as judged by the values which differed considerably. Furthermore, we have not investigated the fracture status of the patient cohort and the American College of Radiology criteria which do not yet include the femur and thoracic spine measurements as shown in our study. In our view there is a need for an adjustment including different measurement locations which is required to make an accurate statement beyond our pilot study presented here.

## 5. Conclusions

The results on bone quality measurements show that in addition to native CT diagnostics, contrast agent administration prior to the measurements can also be used if a compensation factor is established and used to transform the QCT measurements into values which are diagnostically relevant for the diagnosis of osteopenia or osteoporosis. Without having determined the compensation factor of BMD measurements after contrast agent administration, as shown here with a small patient population, CT diagnostics does not yet retain general applicability. Thus, a substantially larger patient population is needed to determine a generally valid or a location-specific factor. It is important to note that CT diagnostics after contrast administration cannot be readily used for BMD determination as outlined by the manufacturer [8], because the agent alters the diagnostically relevant bone densitometry measurements significantly. It is important to emphasize that conversion factors can be used to calculate reliable volumetric BMD measurements for both the spine and hip from routine CT datasets.

## Figures and Tables

**Figure 1 jcm-12-01456-f001:**
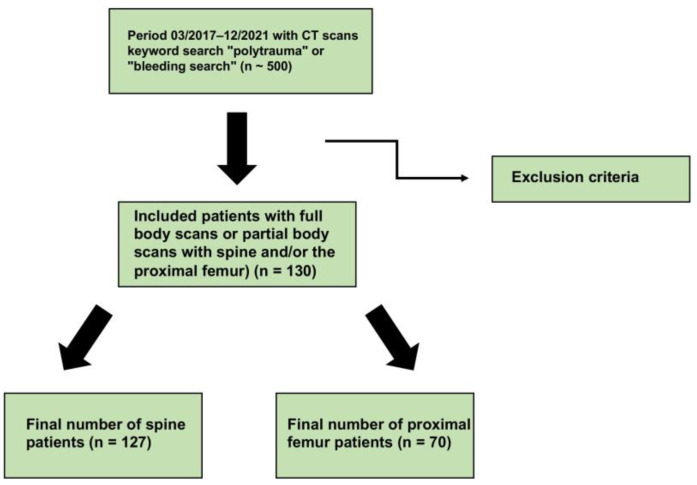
Flowchart about the study overview. After applying the exclusion criteria, 130 patients were integrated into the final examination.

**Figure 2 jcm-12-01456-f002:**
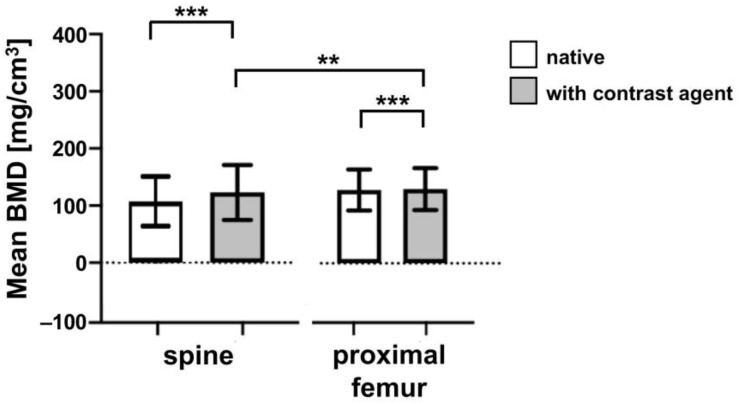
Comparison of bone mineral density (BMD) with and without contrast agent administration. Distribution of BMD of spine patient cohort (n = 127) and proximal femur patients (n = 70); white bars represent the BMD in native CT, gray bars indicate BMD in CT diagnostics after contrast agent application. BMD of patients in mg/cm^3^ shown on the y-axis, localization of BMD shown on the *x*-axis. *** = <0.001; ** = <0.003. Median values are shown.

**Figure 3 jcm-12-01456-f003:**
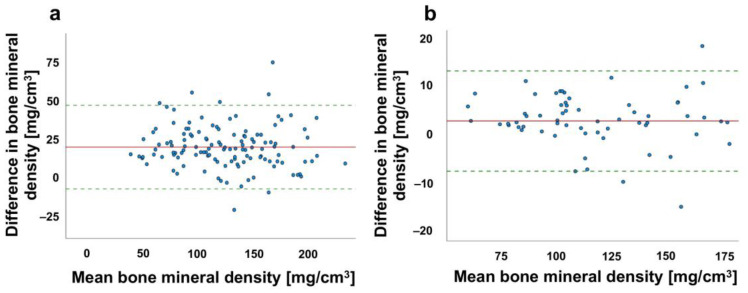
Bland–Altman plot for comparison of bone mineral density (BMD) with and without contrast agent administration. Distribution of BMD of (**a**) spine patient cohort (n = 127) and (**b**) proximal femur patients (n = 70); red line shows the mean value, dashed green lines show the standard deviation (+1.96 and −1.96) and dots represent the distribution of data sets.

**Figure 4 jcm-12-01456-f004:**
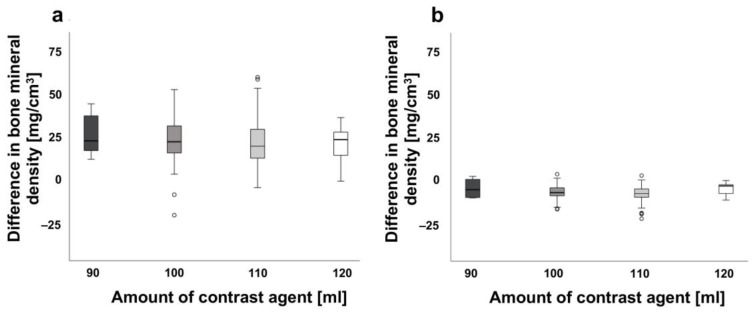
Difference of bone mineral density (BMD) between amount of contrast agents and location. (**a**) Shows the difference of BMD between the amount of contrast agents [mL] (minimum 90 mL to maximum 120 mL) for the spine (n = 127) and (**b**) for the proximal femur (n = 70). Bone mineral density of patients in mg/cm^3^ shown on the y-axis, amount of contrast agents [mL] are presented on the x-axis. Median values are shown. Combined diagram for both sexes.

**Figure 5 jcm-12-01456-f005:**
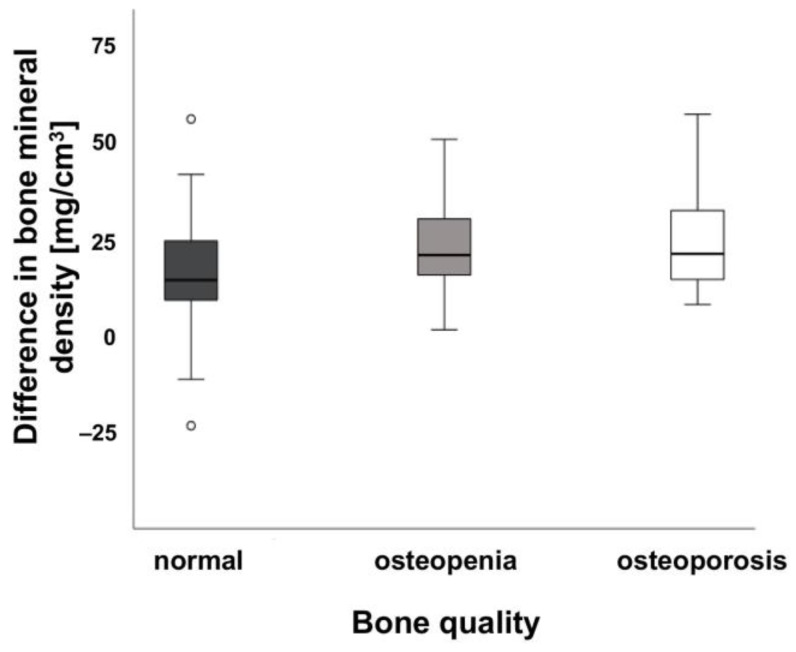
Difference of bone mineral density (BMD) between the bone quality of control subjects (normal), and osteopenia as well as osteoporosis patients. The BMD (mg/cm^3^) is shown on the y-axis, bone quality is presented on the x-axis. Median values are shown. Note the increased BMD in patients.

**Figure 6 jcm-12-01456-f006:**
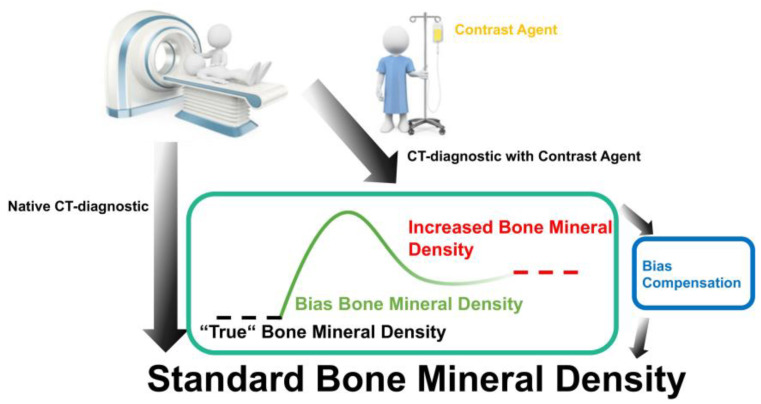
Overview of the experimental setup. The figure clearly shows that a bias compensation is necessary to determine the true bone mineral density.

**Table 1 jcm-12-01456-t001:** Overview of populations data.

Description	Spine (n = 127)	Proximal Femur (n = 70)	*p*
Age mean [years] ± SD	51.23 ± 21.89	57.86 ± 18.63	
**Gender**			
Female [n]	50	31	
Male [n]	77	39	
Mean BMD native [mg/cm^3^] Mean BMD contrast agent [mg/cm^3^]	122.92 ± 48.16 143.80 ± 46.40	123.18 ± 37.48 126.03 ± 36.66	<0.001 (***) <0.001 (***)
**BMD [mg/cm^3^]**			
Amount contrast agent 90 mL	23.56 ± 12.42	4.86 ± 5.22	
Amount contrast agent 100 mL	20.55. ± 16.31	3.06 ± 4.24	
Amount contrast agent 110 mL	18.83 ± 13.31	2.01 ± 6.38	
Amount contrast agent 120 mL	18.01 ± 10.43	4.76 ± 3.44	
Mean BMD [mg/cm^3^] difference	20.88	2.85	<0.003 (**)
Mean Body mass index (BMI) [kg/m^2^]	26.00	26.03	0.020 (*)//0.806 (n.s.)
**Bone quality and BMD [mg/cm^3^]**			
Normal (>120 mg/cm^3^)	23.14 ± 8.23		0.036 (n.s.)
Osteopenia (80–120 mg/cm^3^)	24.94 ± 8.96		0.048 (n.s.)
Osteoporosis (<80 mg/cm^3^)	24.12 ± 10.46		0.067 (n.s.)

Significance level: n.s. = not significant; * = *p* ≤ 0.05; ** = *p* ≤ 0.01; *** = *p* ≤ 0.001.

## Data Availability

The data that support the findings of this study are available on request from the corresponding author. The data are not publicly available due to privacy or ethical restrictions.

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
