# Peer review of "Effect of a Contrast Agent on Bone Mineral Density Measurement in the Spine and Hip Using QCT-Conversion Factor Recommendation"

_jcm, 2023, doi:10.3390/jcm12041456_

Round 1
Reviewer 1 Report
General remarks
There is a lot of interesting and important things in the article, but a full discussion of this does not belong to my duties.
You need to radically rewrite the article after discussing it with a real QCT expert, otherwise you will lower the merit of the journal and lower the merit of QCT.
You need to include Design chart
Authors should be provided with graphs showing direct comparison of BMD values before and after contrast administration. As well as Blend Altamn's diagrams. And also finally decide what kind of statistics they adhere to - parametric or non-parametric and eliminate inconsistencies in the type and content of graphs and values in the text.
If you need contact QCT experts I can try to give you (I can send contact to Editorial team).
My detailed comments are in the attached file.

Author Response
|
Resubmission of the revised manuscript (JCM- 2146725)
“Effect of a contrast agent on bone mineral density measurement in the spine and hip using QCT – Conversion factor recommendation“ by Jäckle et al.
|
Dear Editors,
Thank you very much for having reviewed our manuscript and for the opportunity to resubmit it. As you will see in the point-by-point response below, we have taken the advice of the Reviewers’ comments seriously and have made the according changes in our revised manuscript, provided they were in the scope of our present study. The changes we made are highlighted (in green) in the revised manuscript.
We are very thankful for the comments which helped to improve the presentation of our study.
Kind regards,
- Jäckle
Point-by-point response to the reviewer´s critique, comments and suggestions.
Reviewer #1:
Comments and Suggestions for Authors
General remarks
There is a lot of interesting and important things in the article, but a full discussion of this does not belong to my duties.
We thank the reviewer for the in principle positive assessment.
You need to radically rewrite the article after discussing it with a real QCT expert, otherwise you will lower the merit of the journal and lower the merit of QCT.
We have completely revised the manuscript after consultation with both our QCT experts and statisticians. After having done the respective changes as marked in the revised version of the manuscript, we think that the manuscript is now suitable for publication in the "Journal of Clinical Medicine".
You need to include Design chart
A flowchart (new figure 1) is added to provide a better overview of the study design.
Authors should be provided with graphs showing direct comparison of BMD values before and after contrast administration. As well as Blend Altamn's diagrams. And also finally decide what kind of statistics they adhere to - parametric or non-parametric and eliminate inconsistencies in the type and content of graphs and values in the text.
We discussed the above criticism with our experts. After careful review of the manuscript and the recommendations of our experts and statisticians, and according to the reviewer´s recommendation, we have included Bland-Altman diagrams to provide a better graphical presentation method for the comparison of two measurement methods. (see figure 3)
If you need contact QCT experts I can try to give you (I can send contact to Editorial team).
We thank for the kind offer. However, we have inhouse experts whom we have consulted to improve the revised version of the manuscript according to their expert advice as suggested by the reviewer.
My detailed comments are in the attached file.
21 It is first abbreviation, pleasure correct (BDM) and you need make explanation.
This has been corrected in the revised manuscript.
18 Please note that polytrauma is not a common cause contrast-enhanced CT. As a rule, this patient not old. In general, much more often this is necessary for oncological and other non-traumatic diseases. At the same time, there is a tendency to abandon native CT scans. And this is a problem for the application of opportunistic screening for osteoporosis by QCT without additional correction algorithms. And it is a main reason of this manuscript.
Contrast-enhanced CT is used in our hospital for polytrauma patients in order to visualize and then exclude a possible bleeding as quickly as possible. However, these routine CT diagnostics for polytrauma patients cannot be used in the form of quantitative computed tomography (QCT) for the diagnosis of osteoporosis because according to the company, QCT can only be applied natively. Given the need for better diagnostics for osteoporosis diagnosis, we investigated in our manuscript whether and how contrast agent application can be used for bone densitometry.
45-46 Please state your idea more clearly
We thank the reviewer for pointing this out and have stated our idea more clearly.
53-54-55 This is not entirely true. Positions ISCD in 2015 and then in 2019 approved monoenergetic methods QCT for the diagnosis of osteoporosis (with some comments). For opportunistic research, QCT is acceptable, but its accuracy needs to be improved - this is a fact.
This note of the reviewer is not fully correct! The current "S3 guidelines" on osteoporosis diagnosis and therapy clearly state that a specific diagnosis of osteoporosis requires bone densitometry using DXA measurements as the method of choice because all major therapy studies are based on it. However, it is notable that by now, QCT diagnostics may be superior to the DXA method as a fracture risk predictor according to the guideline group.
See the following link: https://register.awmf.org/assets/guidelines/183-001l_S3_Osteoporose-Prophylaxe-Diagnostik-Therapie_2019-02.pdf
65 Is it really collective? May be collection?
We have changed the heading in “Study design and subjects”.
82-83 Do you really use cushion with synchronous calibration standards? Or you use
asynchronous calibration? You may just check it. If you use asynchronous calibration you need to indicate how often and how you perform it.
We thank the reviewer for this remark, but we used the calibration as described with synchronous calibration standards.
85-86 1) According ISCD position you need perform spine volume BMD measurement at L1-2 level. Thoracic data I think is not valid or you need discuss about it. You may assess effect for Contrast administration. For me it is not clear task.
The reviewer is absolutely correct. According ISCD position the BMD measurement at L1-3 level is necessary. Nevertheless, we wanted to include measurements of the thoracic vertebral bodies in our analysis since data in this region are still very limited. We have added a comment with this regard to the section “limitations” (see page 9; line 371-374).
2) You need measure mean vBMD (from several vertebrae in L1-2 and near – you may include Th12 (11?) L2,3 if L1(2) excluded) data for whole thoracic region had not impossible correlate with normative data like UCSF.
Quantitative parameters were presented showing the mean value, the standard deviation as well as the minimum and maximum values. The Q-Q plot was used to confirm the normal distribution of the data. Measurements with and without the contrast agent were compared using paired T-test. A Bland-Altman plot was used to compare measurements with and without contrast agent administration. Linear regression was calculated to predict the native values from the contrast agent-assisted values. Linear regression was performed separately for each of the spine and proximal femur. A paired T-test was performed to compare BMD measurement differences between the two locations from which the data were obtained. The prerequisite of normal distribution was checked by means of a Q-Q plot. We have added the requested information in the statistics section and Bland-Altman plots (please see new figure 3).
3) How you measure BMD in Hip region?
We very much thank the reviewer for this valuable hint. Analogous to the vertebral body measurement, a range was first defined between the most cranial layer above the highest point of the femoral head and the most caudal layer in a location approx. 1 cm below the trochanter minor. The selected image section was then rotated identically to the vertebral body measurement in all three spatial planes (axial, sagittal, and coronal). A reconstruction of the proximal femur of both volume and surface area was then automatically generated supported by the software. A rectangular region of interest (ROI) was placed in the region of the femoral neck. A blue guide line was then used to mark the most caudal point of the lesser trochanter. The corresponding angle of the ROI was defined using the auxiliary lines. Analogous to the vertebral body measurement, it was performed three times per proximal femur and the mean value from all three measurements was used as the final result in mg/cm3. We have added a sentence to make this important information clear (please see page 2-3; line 86-101).
4) Midway QCT technical reports do include volumetric cortical and trabecular
BMD data, and then they are combined to calculate BMC, which is then divided by area. And this is how the data of the projection IPC are obtained. It is she who is of interest to radiologists and clinicians. Specify why you do not measure the projection BMD for the hip?
After all, it is precisely the calculation of this value that favorably distinguishes KKT midway from other manufacturers - by providing the ability to determine osteoporosis according to the T-criterion based on the projection BMD,
This is absolutely correct and as we described it in the material and methods section. As also shown in the evaluations, we did measure the femur! In fact, as described in the Results and Discussion section, we were also able to detect a statistically significant difference for both the spine (p*** < 0.001) and proximal femur (p*** < 0.001). These data suggest that there is a significant difference between imaging measurements with and without contrast administration in different bones of the body.
5) You need compose additional section with highlight all this principal questions.
We have answered and discussed all of the relevant questions and added them to the revised version of the manuscript.
115 What does it mean «whole spine» ? You need to clarify here or in chapter M&M.
"Whole spine" means the measurements in the thoracic and lumbar spine. We have added this important information as requested.
125-128 The data on the measurement of volumetric BMD for hip is require a very thorough discussion (see lines 85-86). Measurements vBMD taken for spongy or cortical hip?
Etc.
We performed the measurements of the spongy bone in each case, this important information is added in the revised version of the manuscript.
130-131 Oh!!!
1) Pleasure see on box plot – median on the Box Plot is absolutely not
correspond with mean values for spine (see lines 114-115) and not correspond
with mean values for Hip
This is correct because the median is not the same as mean value!!! The median in the box plot was determined just as correctly as the mean value!
2) This plot type not for normal distribution (Line 98) (you marked it some
above).
Boxplots are used to display the distribution of data in a suitable graphical form. Thereby not only the single data are shown, but also their dispersion becomes visible. These representations are particularly well suited for the comparison of two different data sets. Of course, you can use this graphical representation form for both data sets, whether the values are normally distributed or not.
3) by the way: according the box plot data distribution is not normal
All our data sets were statistically checked for normal distribution several times (see 2.3!).
4) Why you make comparison (**): spine vs proximal femur – it is not correct!
The only purpose was to show that there was a statistically significant difference between the two experimental groups, i.e. between the measurements with and without contrast administration.
143-144 Value for proximal femur not correspond with fig 2 (b) where median (-)
In the figure, the median is shown in each case. In the text, we also refer to the mean value. We have clarified this issue to prevent misunderstandings of the reader.
153- 154 You need to consider more beam hardening effects. The volume of injected contrast is related to the body mass index. The body mass index (more precisely, the volume of tissues surrounding the studied object) determines the effect of beam hardening (beam hardening lead to decrease HU). But if you use a synchronous phantom, this effect should in theory be well compensated. This needs to be clarified. In addition, the change in density is important due to the peculiarities of the different vascular structure of the body of the callus and the proximal femur.
The reviewer is absolutely right. Thus, we have added this information in the limitations section. (please see page 9-10; line 369-383)
156 You should change Y annotation for «difference in bone mineral density» For chart b – pleasure correct range for Y axis.
We have modified the labeling of the y-axis as requested. For better comparability of the two diagrams, however, we have left the y-axis range in "b" as it is.
170 This table very bed you need radically correct it
1) Annotation «Baseline characteristics of the populations» not for it
2) Table not correspond with Lines 163-164
3) Column and line labels are not clear!
We are somehow puzzled that our table is not appreciated. However, we have made changes as proposed:
- We have changed the title to "Overview of populations data".
- The reported data in the following sentence "Table 1 summarizes that there is a statistically significant negative association between body mass index (BMI) and the effect on BMD after contrast administration (p* < 0.020)" are now in agreement with the values presented in the table.
- We have changed the outlining of table and hope that it is now clearer.
173-174 Criteria ACR (<80 Ospeoporosis; 80-120 Osteopenia; >120 mg/ml Normal) for lumbar spine and is not applied for femur and thoracic spine
We thank the reviewer for the comment. His/her comments are correct. Our analysis represents indeed the first attempt to classify osteoporosis based on BMD as described here and we address this point in the section “limitations”. (see line 380-383)
199 Green line (Bias bone mineral density) – is very subjective and not conclusive. it's
better to kill it - just two levels before and after.
We think that this line makes the results more pointed and thus, we would prefer if this line remained. Therefore, we have not changed the graphic with this respect.
234 You need to compare your data with some “classic” work where area BMD not statistically affected by contrast administration.
Ziemlewicz T., Maciejewski A., Binkley N. et al. Direct comparison of unenhanced and contrast-enhanced CT for opportunistic proximal femur bone mineral density measurement: Implications for osteoporosis screening // American Journal of Roentgenology. ‒ 2016. ‒ Vol. 206, No 4. ‒ P. 694–698.
We thank the reviewer for this highly valuable advice. We have discussed this work in the discussion section. In addition, we also discuss a most recent review article (Kutleša et al. 2022) which supports our findings. (see page 9; line 327-341)
Reviewer #2:
Comments and Suggestions for Authors
Overall the paper is written well and has great value to the field. However several grammatical and spelling mistakes were found throughout the document which must be addressed. Below are my comments:
We would like to thank the reviewer for the positive evaluation of our manuscript. We have revised the manuscript according to the comments of reviewer #1 and we corrected both errors and linguistic grievances as outlined by reviewer #2.
Line number- Suggested changes Comments
Line 17 Change “of earlier diagnosis” to “for early diagnosis”
The change was made as requested.
18-20 In “the” form of, define “native” when first mentioned
We have added the sentence to …“ can only be applied natively, i.e. without any contrast agent application”. (please see line 19-20).
20 Rewrite to be easy to follow (Ex: “In view of the need for better diagnosis, we tested how contrast agent application could be used for bone densitometry measurements”)
The change was made as requested.
21 Write in full form for BMD the first time and then no need to use full form repeatedly. Check the whole paper. It has been repeated. After this, you can use Just BMD. Same with all acronyms, What is BDM? Should it be BMD?
We have made these changes as requested throughout the entire manuscript. And, of course, BDM should be BMD. Sorry for this typing error.
25 Just BMD
We changed as requested.
26-27 “…which allow for BMD values to be calculated.”
We changed as requested.
27 What is BDM? Should it be BMD?
BDM should be indeed BMD and we have corrected the typing error.
28 No need for ‘the’ BMD, just BMD
We changed as requested.
29 Remove also
We changed as requested.
30 Full form for BMI for first instance, remove “to be determined”.
“..such as weight and corresponding BMI of the patient.”
We changed as requested.
34 Once you mentioned it is a bone disease it indicates that it is a disease of skeleton system. Remove “of the skeleton system”
“all genders” may be more appropriate
We changed as requested.
35-36 “It is characterized by an increased risk of bone fractures, usually due to impaired bone remodelling and the resulting pathological bone microarchitecture”
We changed as requested.
36-39 “This brief characterization is reflected in the Health Consensus Conference definition of osteoporosis [1], which defines osteoporosis as a skeletal disease characterized by decreased bone strength and an increased risk of fracture.”- already mentioned in the previous sentence, Can add this reference to the previous sentence
We have added the reference to the previous sentences.
40 Change “since” to “as”
We changed as requested.
42 “…in the coming decades”
We changed as requested.
43 Change “those” to “these”
We changed as requested.
44 “only about” to “approximately’. Remove “this”
We changed as requested.
46 Remove “additional”
We changed as requested.
47 “…diagnosis of osteoporosis becomes an increasingly important topic for both…”
We changed as requested.
50 BMD
We changed as requested.
51 Osteopenia and osteoporosis (osteoporosis is the severe form of osteopenia. Therefore have the least severe one first)
We changed as requested.
55, 59 BMD
We changed as requested.
53-55 QCT already abbreviated
BMD already abbreviated
We changed as requested.
Content is not clear, how does BMD, DXA and QCT correlate
We have added the sentence: “QCT and DXA diagnostics are different methods to determine BMD.“ To make it more clear to the reader. (see line 57-58)
56 “In this study, we explored how QCT could be used…..”
We changed as requested.
56-57 Reference?
We added the missing reference.
60 Use “CT”, it is already abbreviated in abstract
We changed as requested.
61 Long sentence. Break it. (Ex: This can be compensated….)
We changed as requested.
62 Do not start a sentence with abbreviation. Always use full form, remove “both”
We changed as requested.
63 As you have studied more than one patient, change the wording to “agent in each patient”
We changed as requested.
65 2.1 Better topic (Ex: Study design and subjects)
We changed as requested.
68-69 “Inclusion criteria for the reported retrospective cohort study included 130 patients”
Remove “both”
We changed as requested.
70 Remove “during an”
We changed as requested.
73 Based on what you have selected 70 patients out of 130; Remove comma
Later in the results section n= 127 and 70 for spine and femur respectively (Line 114-116). Clarify this.
In total, there were 130 patients included in the study. Three patients received "only" femur examination and the other 127 received at least spine examination. Maybe this mention is somewhat confusing, but it is a correct statement.
73-74 Rephrase; (Ex: “among them only 3 patients received CT for the proximal femur”
We changed as requested.
74-76 Rephrase in a constructive way (Ex: the native and contrast-enhanced images were taken from the recruited patients at a time interval of no more than six months )
We changed as requested.
80 QCT abbreviation
We changed as requested.
81 In thoracic and lumbar spine region and the proximal femur
We changed as requested.
81-83 Use abbreviations as mentioned earlier. This sentence can move to the introduction
We changed as requested and have moved the sentence to the introduction.
84-86 Isn’t this same as what is mentioned in 79-81
We have deleted the sentence to avoid a repetition.
91 Explain further on selecting amount of contrast medium with body weight
Patients received an amount of 90-120 ml contrast medium via an intravenous catheter, depending on their body weight. The contrast medium was applied via an automatic application system (injector).
94 Remove “of the images”
We changed as requested.
102 Remove “respectively”
We changed as requested.
103 Define “two measurement locations” (Ex: Between spine and femur)
We changed as requested.
106 “The BMD measurement differences”
We changed as requested.
107 “were”
We changed as requested.
107-108 Rewrite (Ex: The tests were two-sided… )
We changed as requested.
110 Rewrite (Ex: Statistical tests were completed with statistics software SPSS or SPSS was used for statistical analysis)
We rewrote as requested.
113 Abbreviation
We changed as requested.
117 Remove “the” and “respectively”, Add table 1 and figure 1 as reference
We changed as requested.
120 Do not start a sentence with abbreviation
We changed as requested.
122 123-Refer table 1
We changed as requested.
125, 127, 136 Check spellings: “femur” not femura
We changed as requested.
126 Remove “respectively”, Add table 1 and figure 1 as reference;
We changed as requested.
Results
3.1 First paragraph-describe the results for thoracic spine and the lumbar spine.
Second paragraph- data for whole spine
Third paragraph- data for proximal femur
We structured the results in the following way: First, description of the comparison between BMD with and without contrast administration, then comparison of contrast administration only comparing to spine and femur... This structuring was not criticized by the first reviewer. Thus, we left our initial structure, because we think it is easier to understand for the reader. However, if this structure is still not acceptable and a different structure is indeed required, we can of course adjust the structure immediately.
Finally, clearly indicate significance between spine and femur with and without contrast administration
The significance levels are indicated, see line 164-166. Note that we refer again to table 1 and figure 2 for better understanding of the results.
Figure 1-Is there a diffidence between male and female as well? Is this a combined graph for both genders?
Fig. 1 (new figure 2) combines the data for both genders, since no gender difference could be identified. To make this point clear, we have added a short paragraph addressing this issue (see page 5; line 212-214).
133 Do not start a sentence with an abbreviation
We changed as requested.
134 Change to “location”
We changed as requested.
137 Use abbreviation
We changed as requested.
138 “both spine and proximal femur measurements”
We changed as requested.
142 Remove however, it has not increased, should be decreased , add reference to table 1
We changed as requested, see below.
Can rewrite in a better way without repeating words (Ex: The mean BMD values for spine were 23.56 ± 12.42 mg/cm3, 20.55 ± 16.31 mg/cm3, 18.83 ± 13.31 mg/cm3 and 18.01 ± 10.43 mg/cm3 for 90,100,110 and 120ml of contrast medium respectively.)
We have made the requested changes.
143-145 Same as above comment. Add reference to table 1
The reference has been added.
147 Add reference to table 1 and figure 2, what are the P values?
Reference and P values have been added.
148 Remove “however”
We removed ”however”.
150 Refer table 1 instead of “see mean values above”
Reference is given to table 1.
151 remove “Here”
“Here” is removed.
154 Change to “…..corresponding distribution…..”
We changed to “…..corresponding distribution…..”.
148-155 Move the whole paragraph to 3.1 section
As described above, we have structured the paragraphs slightly differently.
Figure 2 Having one graph will be easy to compare
We have deliberately chosen the graph to show the difference between spine and femur when placed next to each other. However, if acceptance of the manuscript required this change, we can make the suggested change.
158 Use abbreviation
We changed as requested.
159 Remove “the location of”
“the location of” is removed
159-160 Number of spine patient collective (n = 127) and proximal femur patients (n = 70).
Remove the whole sentence. Instead, add n values in the previous sentence.
(Ex: ….for the spine (n = 127) and (b) for the location of the proximal femur(n = 70) )
We have done the changes as requested.
160 Do not start a sentence with abbreviation, Change to “are shown on the Y axis and the amount of contrast agent (ml) are shown on the x-axis.”
We followed the reviewer´s suggestion.
164 Remove ‘the the’, second ‘BMI’, and ‘the effect on’
We changed as requested.
165 Replace It with This. It is not defined, this refers to just stated. …higher the BMI value, the more contrast…
We changed as requested.
166 ‘in percentage terms’ not needed
We deleted “in percentage terms“ as requested.
166-168 Rewrite sentence
(Ex: …higher body weight corresponds to a higher BMI, resulting in a different distribution of contrast agent.)
We have already modified the sentence as requested by Reviewer #1.
Table 1 Mean BMD native and contrast agent p values are not correct
The values are correct. Note that in the table the values are given for the entire spine and not individually on the thoracic and lumbar spine. This is now clearly stated in manuscript.
P value for BMD native between spine and femur is ns while in femur its 0.003.
P= 0.001 for both locations, with and without contrast agent
This is the result of multiple statistical analysis.
Define “Mean BMD [mg/cm3] difference”.
Is it between native and contrast agent administrated?
If so, the value for spine is not correct. It should be 20.88
The reviewer is absolutely correct. We have corrected this mistake!
Body mass index (BMI) [kg/m2]. If it’s the mean, change to “Mean BMI [kg/m2]”
It is the mean value and we changed it accordingly.
How did you measure “bone quality [mg/cm3]” for spine and femur?
We thank the reviewer for this valuable advice. Bone quality was not measured, but classified on the basis of the BMD values. The table is adjusted accordingly.
Explain significant values, how did you compare?
As described in the "Statistics" section, the differences in bone qualities were compared and evaluated.
172 Section 3.4 Title does not reflect the content
You are absolutely correct, we have amended this issue.
173- 174 Rewrite. Does not make sense
(Ex: …no significant differences in bone quality in terms of BMD when comparing healthy bone, bone with osteopenia or bone with osteoporosis with regard to the administered amount of contrast agent.)
We made the changes as suggested by the reviewer.
How much contrast agent was used?
As described in the material and methods section, we used between 90 ml and 120 ml, depending on the patient´s weight.
How does it correlate to BMD and bone quality?
There was no difference between contrast administration and BMD and bone quality. For this reason, the correlation was not explicitly determined. However, this “deficiency” was explored, and no correlation between the parameters was found. We added this information to the revised manuscript (see page 7; line 259-264).
175-179 Abbreviations.
Reviewer #1 wanted a different label so it was adjusted that way. If this needs to be changed again, please suggest a solution that is acceptable to both reviewers.
Is the difference between native and contrast agent administrated call bone quality?
Explain the process of measuring bone quality
No, this is a misunderstanding. Bone quality is a way to describe bone density-independent (anti-) fracture effects. In general, BMD is considered to be a gauge of bone strength/bone quality and is determined by DXA or QCT diagnostics as described in the manuscript.
Figure 3 Abbreviations.
Please, see above! Reviewer #1 wants it like it is.
How does this correlate to the contrast agent?
See question answered above, there was no direct correlation and this is clearly stated in the revised manuscript.
182 Bone quality “is” presented “on” the x-axis
We changed as requested.
185 Remove “however”
Is removed!
186 Just “Figure 4”. Remove “see”
Is removed!
187 Remove “also”
Is removed!
189 Remove “the”
Is removed!
192, 196 Explain, “mean value after contrast agent application”. Mean value of what?
The mean value after administration of the contrast agent refers to the constant value of 17.33 for the spine and 4.34 for the femur.
194-195 Rephrase to make sense; Explain the values
We have shown the formula with the individual values in line 277-282 (page 7) and then explained it in the following paragraph.
201 Abbreviation
Changes were made as requested.
208 scans , lowercase
We changed as requested.
210 Remove “a”
“a” is removed.
211 Remove “as well”
“as well” is removed.
212 Replace “or” with “of”, remove “instead of or”
Changes were made as requested.
215 What is the outcome of this research in regarding to QCT method and body weight?
Previous studies have shown that QCT method is less affected by body weight.
We did not investigate this question in our study because this issue has been investigated before in other studies and our results would have been confirmatory.
How contrast agent administration, BMI and QCT measurement correlates in this study. Discuss
We found a statistically significant negative correlation between BMI and BMD after contrast administration (p* < 0.020). This means that the higher the BMI value, the more contrast agent is needed per kilogram of body weight. Actually, when using a synchronous phantom, the effect of increased radiation hardening influenced by BMI, density changes due to peculiarities of the different vascular structure of the callus body and the proximal femur should theoretically be well compensated. However, our study suggest that this theoretical consideration is not correct as some of the values differed considerably. We have added a paragraph in the Discussion regarding this issue (see page 9-10; line 374-379).
220 Rephrase (Ex: Osteoporosis primarily affects the elderly and clinical diagnosis is often following a fracture.)
We changed as requested.
225 Abbreviation
We changed as requested.
226 Decades. Then rephrase the next sentence
We changed as requested.
231 Remove “to be considered”
Is removed.
234-235 Remove “in their study” and “the” (“Focusing on vertebral …..”)
Define L1 to L3
Remove “also”
Changes have been made as suggested.
239 Remove “respectively”
Is removed.
246 Remove “has been”
Is removed.
247 Remove “however”
Is removed.
258-259 Grammar, rewrite
Remove “however”
Is removed.
260 Remove “but”, “the” and “the”
“….directly require transformation of measured values…”
Changes were made as requested.
261 Remove “the”
Is removed.
264 “technique” check spellings
We checked the spellings and corrected them
265 “Analyzed”.
Use either American or British throughout the paper
We changed as requested. You are absolutely right! We use now the spelling according to American English!
266 Remove “and”
Is removed.
270-273 Too long. Break
We changed and shorten the sentence as follows: “The results on bone quality measurements show that in addition to native CT diagnostics, contrast agent administration prior to the measurements can also be used if a compensation factor is established and used to transform the QCT measurements into values which are diagnostically relevant for the diagnosis of osteopenia or osteoporosis.“ (see page 10; line 385-388)
274 Explain “the compensation factor BMD measurements”. How do you get this value?
This designation simply means a compensation factor can be determined as shown in this study, and that by using this compensation factor, the BMD can be determined after a contrast agent has been applied.
282 What are your suggestions from this study?
How BMD, contrast agent and BMI co relate? The equation does not explain this
Are there any recommendations of using contrast agents with different body weights?
We did not specifically address this question in our study. However, we plan such a study in the future. However, our current study already shows that contrast agent administration depends in combination with QCT on multiple influencing factors (contrast agent kinetics, influence of age, CT protocol, type and concentration of contrast agent) which need to be addressed. We have added a small paragraph related to these issues in the Limitations section.
We would like to thank the reviewers for their constructive criticism as well as for their many very helpful comments and suggestions. We hope that the corresponding improvements will now allow that our manuscript will now be acceptable for a publication in Journal of Clinical Medicine.
Sincerely,
|
|
|
PD Dr. med. K. B. Jäckle (on behalf of all authors) |

Reviewer 2 Report
Overall the paper is written well and has great value to the field. However several grammatical and spelling mistakes were found throughout the document which must be addressed. Below are my comments:
Line number- Suggested changes Comments
Line 17 Change “of earlier diagnosis” to “for early diagnosis”
18-20 In “the” form of, define “native” when first mentioned
20 Rewrite to be easy to follow (Ex: “In view of the need for better diagnosis, we tested how contrast agent application could be used for bone densitometry measurements”)
21 Write in full form for BMD the first time and then no need to use full form repeatedly. Check the whole paper. It has been repeated. After this, you can use Just BMD. Same with all acronyms, What is BDM? Should it be BMD?
25 Just BMD
26-27 “…which allow for BMD values to be calculated.”
27 What is BDM? Should it be BMD?
28 No need for ‘the’ BMD, just BMD
29 Remove also
30 Full form for BMI for first instance, remove “to be determined”.
“..such as weight and corresponding BMI of the patient.”
34 Once you mentioned it is a bone disease it indicates that it is a disease of skeleton system. Remove “of the skeleton system”
“all genders” may be more appropriate
35-36 “It is characterized by an increased risk of bone fractures, usually due to impaired bone remodelling and the resulting pathological bone microarchitecture”
36-39 “This brief characterization is reflected in the Health Consensus Conference definition of osteoporosis [1], which defines osteoporosis as a skeletal disease characterized by decreased bone strength and an increased risk of fracture.”- already mentioned in the previous sentence, Can add this reference to the previous sentence
40 Change “since” to “as”
42 “…in the coming decades”
43 Change “those” to “these”
44 “only about” to “approximately’. Remove “this”
46 Remove “additional”
47 “…diagnosis of osteoporosis becomes an increasingly important topic for both…”
50 BMD
51 Osteopenia and osteoporosis (osteoporosis is the severe form of osteopenia. Therefore have the least severe one first)
55, 59 BMD
53-55 QCT already abbreviated
BMD already abbreviated
Content is not clear, how does BMD, DXA and QCT correlate
56 “In this study, we explored how QCT could be used…..”
56-57 Reference?
60 Use “CT”, it is already abbreviated in abstract
61 Long sentence. Break it. (Ex: This can be compensated….)
62 Do not start a sentence with abbreviation. Always use full form, remove “both”
63 As you have studied more than one patient, change the wording to “agent in each patient”
65 2.1 Better topic (Ex: Study design and subjects)
68-69 “Inclusion criteria for the reported retrospective cohort study included 130 patients”
Remove “both”
70 Remove “during an”
73 Based on what you have selected 70 patients out of 130; Remove comma
Later in the results section n= 127 and 70 for spine and femur respectively (Line 114-116). Clarify this.
73-74 Rephrase; (Ex: “among them only 3 patients received CT for the proximal femur”
74-76 Rephrase in a constructive way (Ex: the native and contrast-enhanced images were taken from the recruited patients at a time interval of no more than six months )
80 QCT abbreviation
81 In thoracic and lumbar spine region and the proximal femur
81-83 Use abbreviations as mentioned earlier. This sentence can move to the introduction
84-86 Isn’t this same as what is mentioned in 79-81
91 Explain further on selecting amount of contrast medium with body weight
94 Remove “of the images”
102 Remove “respectively”
103 Define “two measurement locations” (Ex: Between spine and femur)
106 “The BMD measurement differences”
107 “were”
107-108 Rewrite (Ex: The tests were two-sided… )
110 Rewrite (Ex: Statistical tests were completed with statistics software SPSS or SPSS was used for statistical analysis)
113 Abbreviation
117 Remove “the” and “respectively”, Add table 1 and figure 1 as reference
120 Do not start a sentence with abbreviation
122 123-Refer table 1
125, 127, 136 Check spellings: “femur” not femura
126 Remove “respectively”, Add table 1 and figure 1 as reference; Results 3.1 First paragraph-describe the results for thoracic spine and the lumbar spine.
Second paragraph- data for whole spine
Third paragraph- data for proximal femur
Finally, clearly indicate significance between spine and femur with and without contrast administration
Figure 1-Is there a diffidence between male and female as well? Is this a combined graph for both genders?
133 Do not start a sentence with an abbreviation
134 Change to “location”
137 Use abbreviation
138 “both spine and proximal femur measurements”
142 Remove however, it has not increased, should be decreased , add reference to table 1
Can rewrite in a better way without repeating words (Ex: The mean BMD values for spine were 23.56 ± 12.42 mg/cm3, 20.55 ± 16.31 mg/cm3, 18.83 ± 13.31 mg/cm3 and 18.01 ± 10.43 mg/cm3 for 90,100,110 and 120ml of contrast medium respectively.)
143-145 Same as above comment. Add reference to table 1
147 Add reference to table 1 and figure 2, what are the P values?
148 Remove “however”
150 Refer table 1 instead of “see mean values above”
151 remove “Here”
154 Change to “…..corresponding distribution…..”
148-155 Move the whole paragraph to 3.1 section
Figure 2 Having one graph will be easy to compare
158 Use abbreviation
159 Remove “the location of”
159-160 Number of spine patient collective (n = 127) and proximal femur patients (n = 70).
Remove the whole sentence. Instead, add n values in the previous sentence.
(Ex: ….for the spine (n = 127) and (b) for the location of the proximal femur(n = 70) )
160 Do not start a sentence with abbreviation, Change to “are shown on the Y axis and the amount of contrast agent (ml) are shown on the x-axis.”
164 Remove ‘the the’, second ‘BMI’, and ‘the effect on’
165 Replace It with This. It is not defined, this refers to just stated. …higher the BMI value, the more contrast…
166 ‘in percentage terms’ not needed
166-168 Rewrite sentence
(Ex: …higher body weight corresponds to a higher BMI, resulting in a different distribution of contrast agent.)
Table 1 Mean BMD native and contrast agent p values are not correct
P value for BMD native between spine and femur is ns while in femur its 0.003.
P= 0.001 for both locations, with and without contrast agent
Define “Mean BMD [mg/cm3] difference”.
Is it between native and contrast agent administrated?
If so, the value for spine is not correct. It should be 20.88
Body mass index (BMI) [kg/m2]. If it’s the mean, change to “Mean BMI [kg/m2]”
How did you measure “bone quality [mg/cm3]” for spine and femur?
Explain significant values, how did you compare?
172 Section 3.4 Title does not reflect the content
173- 174 Rewrite. Does not make sense
(Ex: …no significant differences in bone quality in terms of BMD when comparing healthy bone, bone with osteopenia or bone with osteoporosis with regard to the administered amount of contrast agent.)
How much contrast agent was used?
How does it correlate to BMD and bone quality?
175-179 Abbreviations.
Is the difference between native and contrast agent administrated call bone quality?
Explain the process of measuring bone quality
Figure 3 Abbreviations.
How does this correlate to the contrast agent?
182 Bone quality “is” presented “on” the x-axis
185 Remove “however”
186 Just “Figure 4”. Remove “see”
187 Remove “also”
189 Remove “the”
192, 196 Explain, “mean value after contrast agent application”. Mean value of what?
194-195 Rephrase to make sense; Explain the values
201 Abbreviation
208 scans , lowercase
210 Remove “a”
211 Remove “as well”
212 Replace “or” with “of”, remove “instead of or”
215 What is the outcome of this research in regarding to QCT method and body weight?
Previous studies have shown that QCT method is less affected by body weight.
How contrast agent administration, BMI and QCT measurement correlates in this study. Discuss
220 Rephrase (Ex: Osteoporosis primarily affects the elderly and clinical diagnosis is often following a fracture.)
225 Abbreviation
226 Decades. Then rephrase the next sentence
231 Remove “to be considered”
234-235 Remove “in their study” and “the” (“Focusing on vertebral …..”)
Define L1 to L3
Remove “also”
239 Remove “respectively”
246 Remove “has been”
247 Remove “however”
258-259 Grammar, rewrite
Remove “however”
260 Remove “but”, “the” and “the”
“….directly require transformation of measured values…”
261 Remove “the”
264 “technique” check spellings
265 “Analyzed”.
Use either American or British throughout the paper
266 Remove “and”
270-273 Too long. Break
274 Explain “the compensation factor BMD measurements”. How do you get this value?
282 What are your suggestions from this study?
How BMD, contrast agent and BMI co relate? The equation does not explain this
Are there any recommendations of using contrast agents with different body weights?
EXTENSIVE GRAMMAR CHECK, SPELLING CHECK AND ABBREVATION CHECK NEEDED
Author Response

(The authors gave the same response as above.)

Round 2
Reviewer 1 Report
Dear colleagues you made a great job. Im just try to help and I'm very -very appreciate for your attention to my remarks.
After adjusting a small additional number of comments, you can send it to print. Unfortunately, I did not have time to see how the comments of another reviewer were corrected. But I am sure that the authors did everything correctly.
I did not comment on those points where I agree.
Please see one critical remark to Figure 2 and some minors. You can see below.

Author Response
|
Journal of Clinical Medicine- 2146725 Resubmission of our revised manuscript “Effect of a contrast agent on bone mineral density measurement in the spine and hip using QCT – Conversion factor recommendation“ by Jäckle et al.
|
Dear editors,
Thank you very much for the quick and thorough review of our revised manuscript. Reviewer #2 was pleased with our changes and had no additional suggestions for improving the manuscript. Reviewer #1 was overall positive and had a valuable suggestion, which we address in our response below.
We are very grateful for having had the opportunity to resubmit our manuscript, following the advice/suggestions of Reviewer #1. We agree that his/her comment helped to further improve our manuscript.
As you will see, our response (in blue) explains how we addressed Reviewer #1 comments. The changes made in the revised manuscript are highlighted in green.
Kind regards,
- Jäckle
Response to the reviewer´s critique, comment and suggestion.
Reviewer #1:
Dear colleagues you made a great job. I’m just try to help and I'm very -very appreciate for your attention to my remarks.
After adjusting a small additional number of comments, you can send it to print. Unfortunately, I did not have time to see how the comments of another reviewer were corrected. But I am sure that the authors did everything correctly.
I did not comment on those points where I agree.
Please see one critical remark to Figure 2 and some minors. You can see below.
We would like to thank the reviewer for the overall positive feedback regarding our revision. We are extremely grateful for his/her assistance and the extremely helpful comments, which have significantly improved our manuscript. Concerning the comment referring to the second reviewer: Yes, we did everything he/she requested and Reviewer #2 had no additional suggestions for a further improvement of the manuscript.
Language
Pleasure check terms “spongy BMD” (may be “Trabecular BMD”) “patient collective” (may be “Patient cohort”)?
We have changed as requested.
53-54-55
53-54-55 This is not entirely true. etc. Link in Germany, I am non well in this. My opinion not changed but remark not critical.
Giving the link in German, that was stupid, sorry! I addressed this point in the last revision. According to current guidelines, it is indeed the case that osteoporosis guidelines and treatment recommendations are mainly based on DXA diagnostics. Although QCT examination is not considered equivalent to DXA, some earlier studies have shown a significant correlation between DXA and BMD determined by QCT.
The paragraph is therefore correct in terms of the statement and has therefore been left unchanged
85-86
Number 4
4) Midway QCT technical reports do include volumetric cortical and trabecular etc.
I think we understood each other partially. But this can be left
We thank the reviewer and agree.
About Figure 2
I plotted the histogram according manuscript text data (lines 152-153 for spine, 163-164 for hip). Then I fused your and my plots for demonstrate data discordance and please – follow see my reasoning: see fig N below and in additional JPG file:
Why this is absolutely impossible situation? Please e.g. for spine:
We agree: it is not an impossible situation. Actually, we never mentioned that it would be. We took the reviewers critical comment seriously and provide a new Figure as suggested by the reviewer.
1) Median BMD spine before contrast enhancement (CA) approx. 25 mg/ml (!). This is very strange because it is deep osteoporosis level (much low 80 mg/ml). But your patients were not old 55.12 ± 19.24 y.o.)
Mean BMD spine before CA 129 mg/ml ; D (mean-median) ≈ 100 mg/ml (see blue arrow!).
But in normal distribution Median ≈ Mean.
2) [75% percentile] before CA ≈ mean value before CA – it is very strange!
[25-75 percentile] before CA ≈ 130 mg/ml from your plot
[25-75 percentile] after CA ≈ 60 mg/ml from your plot
In practical it is unreal situation – this data must be very similar, just some high after CA.
For the hip joint, too, strange data for Figure 2, but this oddity is less obvious.
First of all, we would like to thank the reviewer for going through so much trouble to enrich our manuscript. We discussed the “problem” again with our statisticians to solve the apparent discrepancy:
The reason is that we do not show the natural BMD values, but the difference in BMD readings between measurements after contrast agent administration and those without the contrast agent administration. We apologize for having created this misunderstanding and the associated confusion. As a consequence, we have now explicitly pointed this out in our manuscript in order to avoid a misunderstanding by the reader. (see below)
Fig N
My strong suggestion for you is to present the data not in boxplot format: Median (25th, 75th percentile, max-min). Because this is more acceptable for non normal distribution.
But you showed earlier that the data is normally distributed (lines 133-134). Therefore, you should not use box plots (box with whiskers), but simpler histograms (mean +/- error of mean) or (mean +/- STD). See my plots on fig N.
It's also important to recheck your raw data and plots again for peace of mind.
Perhaps the level of patients and the level of the vertebrae are mixed. Perhaps there is another explanation.
I had to be consulted our statistic expert and she was agreeing with me!
We have again analyzed the data and they were correct. As already explained above, these values are now presented in our new Figure 2 and they represent the mean values of the differences between BMD with and without contrast medium administration. We have added this to the manuscript and changed the graph as requested.
170 This table very bed you need radically correct it
I’m not full agree but OK
We thank the reviewer for the OK. Thus, we have left the table as it was since there were no specific suggestions of how to improve the table. However, if this reviewer still does not like our design, we can make changes as he/she requests them.
Figure 5
Please correct Y axis name as figure 4
In the caption under the figure, you correctly indicated «Difference of bone mineral density (BMD)», but if it be possible, correct Y axis name.
Thank you
Sincerely
We have made the changes as requested!
Reviewer #2:
The paper has improved considerably after implementing the changes.
We thank the reviewer for his/her positive feedback.
We would like to thank the reviewers again for their very constructive criticism including their very helpful comments and suggestions which clearly improved our manuscript. We hope that our revised version of the manuscript is now suitable for publication in Journal of Clinical Medicine.
Sincerely,
|
|
|
PD Dr. med. K. B. Jäckle (on behalf of all authors) |

Reviewer 2 Report
The paper has improved considerably after implementing the changes.
Author Response

(The authors gave the same response as above.)
